

# Can harbor seals (*Phoca vitulina*) discriminate familiar conspecific calls after long periods of separation?

Mila Varola[1,2,*], Laura Verga[1,3,*], Marlene Gunda Ursel Sroka[2,4], Stella Villanueva[2], Isabelle Charrier[5] and Andrea Ravignani[1,2]

[1] Comparative Bioacoustics Research Group, Max Planck Institute for Psycholinguistics, Nijmegen, The Netherlands
[2] Research Department, Sealcentre Pieterburen, Pieterburen, the Netherlands
[3] Department of Neuropsychology and Psychopharmacology, Maastricht University, Maastricht, The Netherlands
[4] Department of Behavioral Biology, University of Münster, Münster, Germany
[5] Paris Saclay Institute of Neuroscience, Université Paris-Saclay, Orsay, France
[*] These authors contributed equally to this work.

Corresponding authors
Laura Verga, laura.verga@mpi.nl, lv.lauraverga@gmail.com
Andrea Ravignani, andrea.ravignani@mpi.nl

## ABSTRACT

The ability to discriminate between familiar and unfamiliar calls may play a key role in pinnipeds' communication and survival, as in the case of mother-pup interactions. Vocal discrimination abilities have been suggested to be more developed in pinniped species with the highest selective pressure such as the otariids; yet, in some group-living phocids, such as harbor seals (*Phoca vitulina*), mothers are also able to recognize their pup's voice. Conspecifics' vocal recognition in pups has never been investigated; however, the repeated interaction occurring between pups within the breeding season suggests that long-term vocal discrimination may occur. Here we explored this hypothesis by presenting three rehabilitated seal pups with playbacks of vocalizations from unfamiliar or familiar pups. It is uncommon for seals to come into rehabilitation for a second time in their lifespan, and this study took advantage of these rare cases. A simple visual inspection of the data plots seemed to show more reactions, and of longer duration, in response to familiar as compared to unfamiliar playbacks in two out of three pups. However, statistical analyses revealed no significant difference between the experimental conditions. We also found no significant asymmetry in orientation (left *vs.* right) towards familiar and unfamiliar sounds. While statistics do not support the hypothesis of an established ability to discriminate familiar vocalizations from unfamiliar ones in harbor seal pups, further investigations with a larger sample size are needed to confirm or refute this hypothesis.

# INTRODUCTION

## Long-term recognition in pinnipeds

Long-term recognition, which incorporates the storage and subsequent retrieval of information, plays an active role in human and animal behavior and communication.

Indeed, it may be key for the survival of a species to recognize items stored in memory, such as sounds or images, as advantageous or potentially harmful (*e.g.*, *Mendl et al., 2001*). Within animal species, recognition is further believed to enhance the individuals' fitness, playing an active role in mate reunion, parental care, and territory control (*Insley, 2000*). Like many other animals, pinnipeds consistently rely on long-term memory to communicate and to interact with their environment and several species show impressive long-term memory capacities for visual information, problem-solving strategies, and vocal recognition (*McCulloch, Pomeroy & Slater, 1999*; *Reichmuth Kastak & Schusterman, 2002*; *Pitcher, Harcourt & Charrier, 2010*). The ability to recognize target individuals in the wild can provide a strong advantage in the context of high natural selective pressure, as for example in mother-pup recognition (*Charrier, 2020*). However, long-term vocal recognition in pinnipeds has been suggested as an evolutionary by-product of strong maternal imprinting experienced during the rearing period rather than an adapted trait (*Charrier, Mathevon & Jouventin, 2003*). Research on northern fur seals (*Callorhinus ursinus*) showed that females and their pups were able to respond to each other's vocalizations even several years after weaning (*Insley, 2000*). While other studies have demonstrated similar long-term memory in other otariid species (*Charrier, Mathevon & Jouventin, 2003*; *Pitcher, Harcourt & Charrier, 2010*), comparable work in phocids is lacking. A possible reason for this deficiency may depend on differences in both reproductive and maternal strategies between the two pinniped families: otariids form moderate to high density breeding colonies, in which females frequently separate from their pups to forage at sea throughout the entire lactation; instead, most phocids are not colonial and females stay with their young until weaning (*Hammill et al., 1991*). These different strategies may impact mother-pup recognition, which has been suggested to be less needed in phocids than in otariids (*Job, Boness & Francis, 1995*; *McCulloch, Pomeroy & Slater, 1999*; *McCulloch & Boness, 2000*). Field observation suggests that recognition abilities may exist in phocid pups (*Renouf, Lawson & Gaborko, 1983*; *Lawson & Renouf, 1987*; *Insley et al. , 2003*), but it has never been experimentally investigated.

## Harbor seal—a study model

Harbor seals (*Phoca vitulina*) have a unique maternal strategy among phocids as females perform short foraging trips during lactation (*Boness & Bowen, 1996*) while pups show high mobility and a tendency to gather during the breeding season (*Sullivan, 1982*; *Sauvé, Beauplet & Hammill, 2015a*). This maternal strategy implies frequent mother-pup separations, possibly requiring a strong need for an individual vocal recognition. Indeed, several preliminary studies have shown that pup vocalizations (known as mother-attraction calls, MACs) are individually distinctive by their fundamental frequencies, pup age, and sex, suggesting this as a key factor for pup recognition and for the maintenance of mother-offspring contact (*Renouf, 1984*; *Perry & Renouf, 1988*; *Khan, Markowitz & McCowan, 2006*). Eventually, in 2015, Sauvé and colleagues demonstrated that the MACs of harbor seal pups contain individual vocal signatures (*Sauvé, Beauplet & Hammill, 2015b*), and that harbor seal mothers can recognize and distinguish the calls of their own pup by memorizing the pup call signature (*Sauvé, Beauplet & Hammill, 2015a*). However, similar evidence in

pups is lacking as harbor seal mothers do not vocalize, and whether they possess such ability is yet to be determined with other conspecifics' calls.

As harbor seal pups spend the rearing period in close contact with other pups, they show their most social behavior during this time. This close social proximity exposes them to conspecific MACs from an early stage of life (*Sullivan, 1982*; *Bowen, 1991*) and could drive the development of an individual recognition between familiar individuals. Such social recognition might reduce aggressive interactions towards familiar individuals (*Tripovich et al., 2008*), as shown in grey seal weaned pups (*Robinson et al., 2015*). In turn, the reduction of aggressive behaviors may limit the pups' energy expenditure and may act as a driver for sociality and for the development of altruistic behavior (*Axelrod & Hamilton, 1981*). The frequent social interactions might be facilitated by the strong site fidelity shown by harbor seals throughout the year (*Cordes & Thompson, 2015*), in turn promoting recognition of familiar individuals via acoustic, visual, and olfactory cues (*Robinson et al., 2015*). However, memorization of conspecific vocalizations may not be a functional target per se, but instead constitute a by-product of other potential memory capacities. Conversely, one could hypothesize that harbor seal pups would not memorize conspecifics' voices, either because of their young age or because this would not bring any advantage to their own survival.

As a first, exploratory study, we aimed at testing whether it is possible for harbor seal pups to discriminate between MACs from familiar and unfamiliar pups even after a long separation. To this aim, we took advantage of the rare occasions in which already rehabilitated pups were admitted into rehabilitation for a second time in their lifespan during their post weaning period. During this second period, we presented three wild seal weaners with a playback sequence composed of familiar and unfamiliar MACs and measured each animal's orientation and behavioral response to these calls.

## MATERIALS & METHODS

### Study site and animals

This research was performed at the Sealcentre Pieterburen, a seal rescue and rehabilitation center in the Netherlands, which rehabilitates more than 150 phocids every year. The animals reached the Sealcentre as pups in summer 2018 (cause: maternal separation). After a successful first rehabilitation, three individuals returned to the center due to parasitic pneumonia, a relatively common condition seen in young harbor seal weaners (winter 2018–2019). The three seals were 7, 8, and 12 months old at the moment of the experiment, respectively (see Table 1). During their first stay in rehabilitation, pups were housed with at least one companion and consistently exposed to their companions' naturally emitted MACs. While harbor seals usually emit a considerable number of MACs as pups, they gradually stop vocalizing towards the end of the weaning period, which occurs when they are 26–42 days old (*Bowen, 1991*). Furthermore, they predominantly become solitary as weaners (*Bowen, 1991*; *Biolsi, 2017*). Social interactions between conspecific pups progressively increase with age, as it has been shown for social play behavior (*Renouf & Lawson, 1987*). Since the subjects of the present study were released in the wild after the

**Table 1 Characteristics of the tested seals.** The first column shows the name and the sex of the three tested seals. The following three columns respectively refer to: the date of the first admission (cause: maternal separation), the estimated age at the first admission, and the age and sex of its companion(s). The last columns show how much time the companions spent together, how much time passed between their meeting and the experiment, and how much time passed between the test and the pups' separation (time units are expressed in days). For seal-1, the details of each of the four companions (c1, c2, c3, c4) are provided separately

| Seal ID and sex | First arrival - Estimated age | Age and sex of companion(s) | Test date - Estimated age | Days with companion(s) | $\Delta$Time (test-meeting) | $\Delta$Time (test-separation) |
|---|---|---|---|---|---|---|
| seal-1 M | 26/06/2018 - 7–10 days | all companions: 7–10 days F | 03/01/2019 – ca 7 months | c1: 22 c2: 11 c3: 23 c4: 34 | c1: 190 c2: 191 c3: 191 c4: 191 | c1: 168 c2: 180 c3: 168 c4: 157 |
| seal-2 M | 22/05/2018 - 5–6 days (premature) | 3–4 days (premature) M | 10/03/2019 - ca 8 months | 45 | 291 | 246 |
| seal-3 F | 24/06/2018 - 7–10 days | 7–10 days F | 16/06/2019 - ca 12 months | 48 | 355 | 307 |

end of the weaning season, it is highly likely that they were not exposed to further MACs from any conspecific pup.

The first seal tested (seal-1) was a male, approximately 7–10 days old upon first arrival at the Sealcentre (Table 1). Seal-1 was housed during rehabilitation with several other pups, all females of the same age. The other two tested seals (seal-2 and seal-3) were each housed with only one companion. Seal-2 and its companion, both males, arrived at the Sealcentre as premature pups (5–6 and 3–4 days old, respectively). Seal-3 and its companion, both females, were about 7–10 days old at arrival. Age was determined by experienced veterinarians basing on the presence and appearance of the umbilical cord.

## Sound recordings

Aerial vocalizations were recorded from each seal admitted at the Sealcentre during the health assessment upon intake, which occurred in summer 2017 and summer 2018. Twelve wild-born Eastern Atlantic harbor seals (*Phoca vitulina vitulina*) were recorded in 2017 and 2018 to build playback stimuli and three were tested in 2018 and 2019 with these playbacks. Recordings were taken with a Sennheiser ME-66 unidirectional microphone (frequency response of 40–20,000 Hz $\pm$ 2,5 dB; Sennheiser electronic GmbH & Co. KG, Wedemark, Germany) connected to a Zoom H6 digital recorder (Zoom Corporation, Tokyo, Japan). The sampling frequency of all recordings was 48 kHz with 24-bit quantization, and they were saved as WAV files. An MZW-66 foam windshield was used to cover the microphone and protect it from water splashes. All pups admitted to rehabilitation were initially quarantined in pairs or groups, in acoustic contact with each other. Playback consisted of vocalizations recorded from MACs emitted by the test subjects' former quarantine companion(s) during summer 2018 (familiar MACs), and MACs from unknown pups recorded during summer 2017 (unfamiliar MACs).

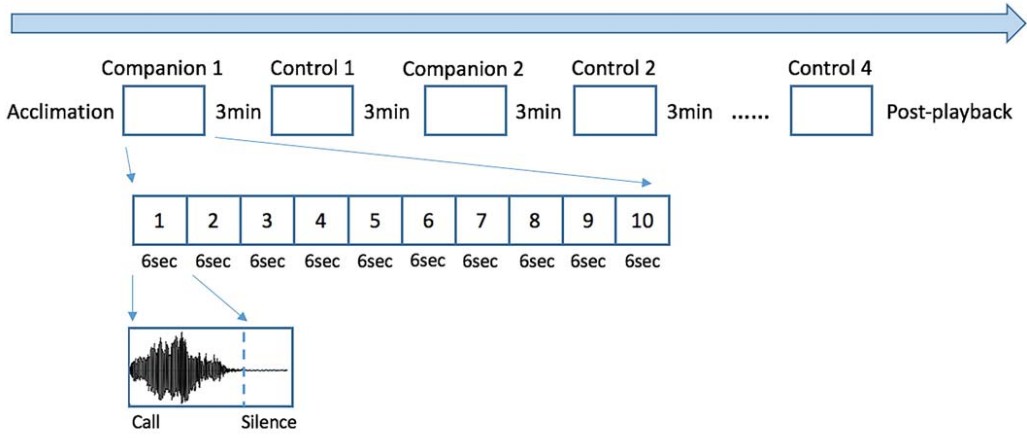

**Figure 1 Experimental design.** The playback consisted of 8 blocks, each one including a sequence of 10 identical calls. Four of the blocks contained calls belonging to the former companions of the tested seal, while the remaining four consisted of unfamiliar pups' calls to serve as controls. Every call unit lasted 6 seconds and between every block there were 3 minutes of silence. Silent periods were added at the beginning and end of the playback ("acclimation" and "post-playback"). The order of the blocks was randomized and was different for each tested seal.

## Playback procedure

Each seal-specific playback was created using Praat 6.0.11 and consisted of a sequence of 8 blocks, each one containing 10 identical calls which onsets occurred every 6 seconds (Fig. 1. See also *Ravignani, 2019*). Half of the blocks consisted of familiar vocalizations ("companion"), while the other half consisted of unfamiliar vocalizations ("control"). To avoid possible habituation to sounds (*Perry & Renouf, 1988*), blocks were separated by 3 min of silence. In addition, a silent period of minimum 60s was added before ("acclimation") and after ("post-playback") the playback presentation, to let the seal habituate to the new situation and to rest after the experiment. Since seal-1 shared its quarantine enclosure with several other pups, for the familiar vocalizations we selected calls from four companions; calls from each companion were grouped within a block and the order of blocks was randomized. The other two tested seals (seal-2 and seal-3) each shared their quarantine with only one companion; therefore, the selected familiar vocalizations belonged to their respective companions. For each test subject, the seals used as control were of the same sex and age as the companion(s). The alternation between companion and control blocks was randomized and presented in a different order to each seal (see Fig. 1). Due to a technical problem, for one of the seals tested (seal-3), the first block of calls contained only 9 calls instead of 10.

## Apparatus and experimental procedure

Each harbor seal, once successfully rehabilitated and in good health condition, was exposed to a playback on the release day. The experiment was performed while the seal was in a box used to release the animal to prevent visual distractions. Particular care was taken to avoid external sources of noise during testing. Since rehabilitated seals were familiar with being transferred and weighed in tightly fitting boxes (53 × 123 × 60 cm), the effect

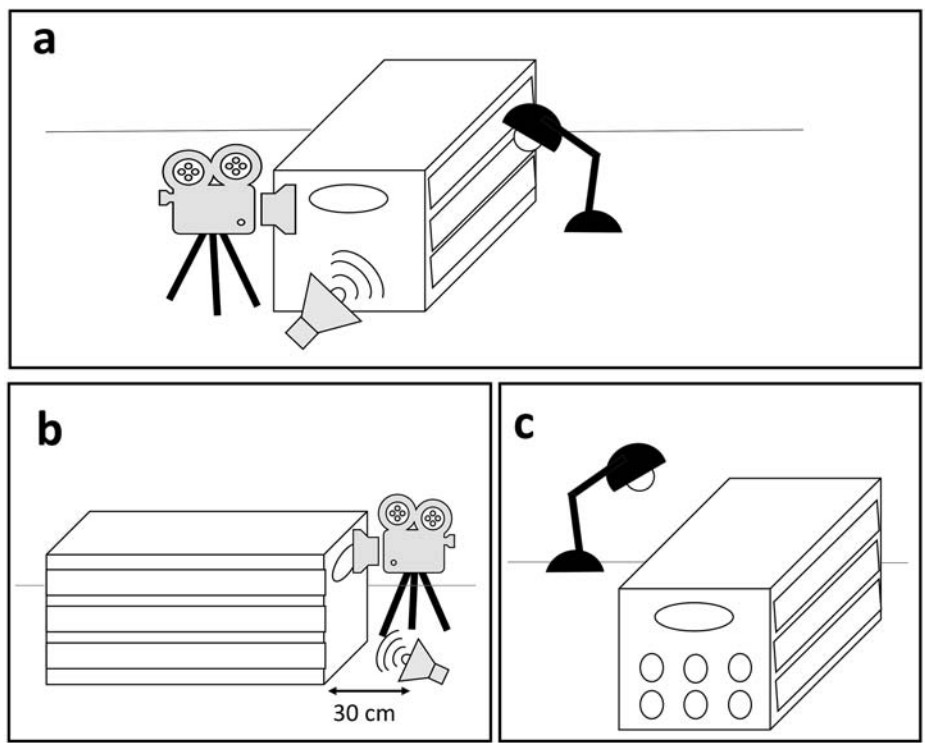

**Figure 2** **Overview of the experimental area and set-up.** (A) The experiment area is isolated so that the seal is not distracted by external visual stimuli. (B) The camera is placed behind the seal and the speaker is placed 30 cm from the box. (C) Additional light is provided by a lamp, to guarantee a clear image.

of short periods of interaction with humans and confinement on the behavioral response was assumed to be negligible (*Osinga & Hart, 2010*). The experiment was recorded with a Zoom Q8 camera, recording at 25 fps from the back side of the animal (*i.e.*, behind its back flippers). The playback was broadcasted from a JBL Flip 2 speaker (frequency response 100 Hz–20 kHz), connected via cable to an iPhone 5S. The speaker was placed within a distance of 30 cm from the back of the box (see details below) and the sound pressure level was adjusted to simulate another seal pup located approximately 1–2 m from the experimental subject. SPLs varied for each call to maintain their original natural amplitude at 1–2 m, averaging 80 dB (measured with a CHECK MATE CM-130 sound level meter). Only after the equipment was in place and working, the playback was triggered remotely with the experimenter out of sight (Fig. 2).

## Behavioral assessment

Behavioral analyses and playback annotations were performed using BORIS (Behavioral Observation Research Interactive Software) v. 7.1.3 - 2018-11-16 (*Friard & Gamba, 2016*). The anatomy of harbor seals affords a wide dynamic visual field; specifically, a seal performing eye movements can reach a visual field extending to 121° on the dorsal side and 210° in the horizontal plane (see *Hanke, Römer & Dehnhardt, 2006* and figures therein); however, the dimensions of the experimental box only allowed the seals to turn their eyes

or heads towards the sound source. Hence, we hypothesized that if seals could discriminate between familiar and unfamiliar voices, they would react to the broadcasted vocalizations by turning their heads towards the sound source placed behind them. We scored the behavioral reaction with a commonly used assay known as the orientation response (*Insley, 2000*; Fig. 3). To avoid any possible bias, all annotations were performed on the video without sound, and the annotator was blind to the experimental condition. Moreover, during sound playback experiments, scientists often noticed asymmetries in orientation (left *vs.* right) towards familiar and unfamiliar sounds in animals (*Ghazanfar, Smith-Rohrberg & Hauser, 2001*; *Böye, Güntürkün & Vauclair, 2005*; *Siniscalchi et al., 2012*). The brain's left hemisphere controls vocal production and perception, and the recognition of a familiar sound (usually a conspecific call) generally corresponds to a predominant turn to the right side. Thus, we hypothesized that the seals would show different orientation parameters depending on the familiarity of the acoustic stimulus. We also hypothesized that a difference in the reaction to the sounds might be reflected in the look duration and in the time elapsed between playback onset and seal's first look to the sound source (latency). For example, a faster response (*i.e.*, a shorter latency) and a longer look might be associated with a familiar vocalization. Thus, we measured three dependent variables: (1) number of looks, defined as either head turning at a >90° angle in the horizontal or dorsal plane or as an eye presentation to the camera; (2) duration of each look; (3) latency. Responses to individual calls within a block (duration = 60 ± .001 s) were summed (number of looks) or averaged (duration, latency) to provide a base for the analyses (see Table 2). This way, the analyses would show the seal's reaction to the type of calls overall, either familiar (companion) or unfamiliar (control). Only reactions that started after the beginning of the block were considered. Vocal responses were not uttered in the current experiment. For latency and duration, we conducted the analyses twice: first, we analyzed only durations and latencies corresponding to looks. This method led to a removal of data from 3 out of 24 blocks (2 companion blocks from seal-1, 1 control from seal-3) in which no looks occurred; therefore, we conducted the analysis again by including these blocks and inputting a minimal duration (0s) and maximum latency (60s). As there was no difference in the outcome of the statistical analyses, we report here only results based on latency and duration corresponding to looks.

## Statistical analyses

All analyses were performed with RStudio version 1.3.959 (*R Studio Team, 2020*) running R version 4.0.2 (*R Core Team, 2020*). Results were deemed significant at an alpha level of .05. Fisher's exact tests (package *stats*) were performed on contingencies tables (one per seal) pitting the direction of reaction (left *vs.* right) against sound familiarity (companion *vs.* control) to test for possible turning asymmetries in response to familiar or unfamiliar vocalizations. In addition, pairwise-correlations (Spearman's rank coefficient) were conducted to evaluate the relation between dependent variables.

We employed generalized linear mixed models (GLMM; package *lme4*; *Bates et al., 2015*) to model the seals' reactions to the playback blocks. After checking for data dispersion (package *blmeco*) and based on Shapiro-Wilk tests and on visual inspection of boxplots,

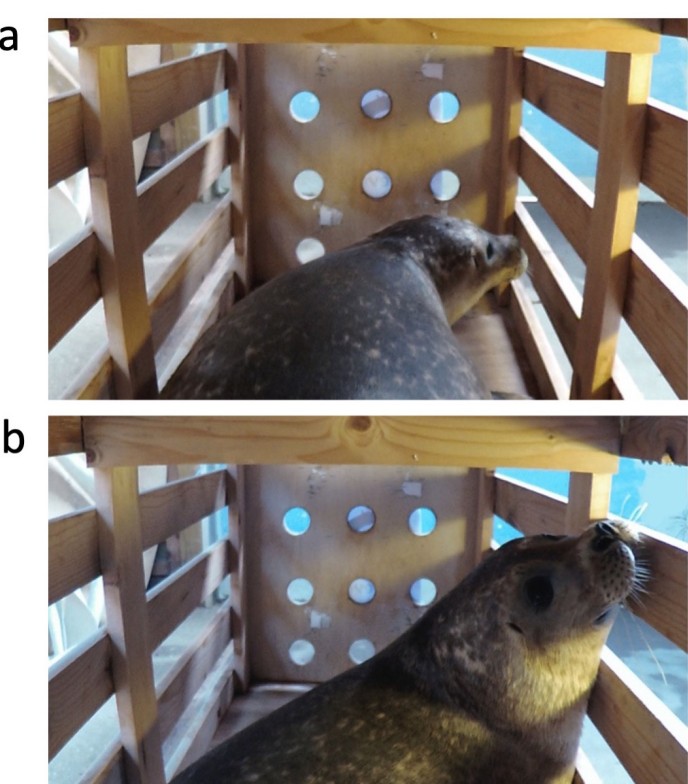

**Figure 3 Representation of the recorded behavioral response.** (A) Eye presentation, the seal turns its head at an angle of < 90°; (B) head turn, the seal rotates its head towards the sound source at an angle of > 90°.

histograms, qq-plots, and Cullen and Frey graph (package *fitdistrplus*) we concluded that the number of looks could be best approximated by a Poisson distribution (overdispersion parameter <1.4, package *blmeco*), while duration and latency were better described by Gamma distributions. For all three dependent variables, the statistical models included *familiarity condition* (companion *vs.* control) as fixed effect and *seal ID* as random effect. Details of the employed models are summarized in Table 3. Furthermore, we used principal component analysis (PCA; packages *stats, factoextra*) to transform the three response variables (number of looks, duration, latency) into a single global measure describing the strength of response to playback (*McGregor, 1992*; see also *Vehrencamp et al., 2003*; *Sauvé, Beauplet & Hammill, 2015b*). Principal component (PC) scores were computed for PCs with eigenvalues greater than 1. PC scores were fed into a linear mixed effects model with *familiarity condition* as fixed effect and *seal ID* as random effect.

For all mixed models, model fit was reported as coefficients of determination ($R^2$) calculated based on (*Nakagawa, Johnson & Schielzeth, 2017*) (package *performance*); however, we caution on the interpretability of these coefficients for generalized linear models (*Sotirchos, Fitzgerald & Crainiceanu, 2019*; *Piepho, 2019*).

**Table 2 Descriptive summary of individual seals' responses to blocks.** Responses are reported as total count of looks, their average latency and duration, for all conditions, companions (abbreviation: comp), and controls (abbreviation: cont). Standard deviations are reported in brackets.

| Seal ID | | Number of Looks | | Latency | | Duration | |
|---|---|---|---|---|---|---|---|
| seal-1 | 13 | comp:4 | 26.37 (19.20) | comp:22.29(16.87) | 5.17 (3.00) | comp:5.04 (0.39) | |
| | | cont:9 | | cont:28.41 (22.43) | | cont:5.23 (3.86) | |
| seal-2 | 35 | comp:22 | 17.80 (15.21) | comp:13.32 (12.31) | 9.61 (5.66) | comp:11.56 (6.45) | |
| | | cont:13 | | cont:22.28 (18.30) | | cont:7.66 (4.80) | |
| seal-3 | 38 | comp:21 | 6.64 (6.38) | comp:7.97 (7.53) | 11.66 (9.23) | comp:14.75 (12.93) | |
| | | cont:17 | | cont:5.31 (5.80) | | cont:8.57 (2.52) | |

### Ethics approval

All focal individuals were kept in the Sealcentre Pieterburen, one of the biggest seal rehabilitation centers in the Netherlands, for clinical reasons only. Data collection was non-invasive and adhered to the guidelines of the Association for the Study of Animal Behavior (*Buchanan et al., 2012*). All procedures were approved by the Sealcentre veterinarians. The animals were neither captured nor kept longer than necessary to run this study. The present study involved behavioral testing only and did not cause any distress or pain to the animals. After the study, the animals were released into the wild according to the regulations and protocols of the Sealcentre Pieterburen.

## RESULTS

### Descriptive analysis

A summary of response patterns to playbacks is presented in Table 2 and Fig. 4. Visual inspection of the data suggested that seal-1 seemed to look less often than the other two seals towards the sound source. In addition, seal-1 looked less often to companion blocks than controls, while the opposite pattern (more looks to companions relative to controls) was observed in seal-2 and seal-3. Still judging from visual inspection, latencies were longer in control blocks in seal-1 and seal-2. Duration of reactions visually resembled the pattern observed for the number of looks, with seal-1 showing looks of similar length for control and companion blocks, and seal-2 and seal-3 displaying longer looks towards companions.

Because seal-1 was exposed to calls from multiple companions, we looked more in depth whether it reacted differently to any of them (Fig. 5). Seal-1 did not react at all to two companions, while the remaining two companions received 2 looks each. Controls received either 1 look, 3 looks, or 4 looks. For companions, durations were similar, while they were more variable for controls. Latencies were variable for all blocks.

### Lateralization of response and correlations

For each seal, there was no statistical association between the familiarity of the sound and the side of the orienting response (Fisher's exact tests; seal-1: $OR = .44$, $p = 1$; seal-2: $OR = 1.39$, $p = .73$; seal-3: $OR = .29$, $p = .10$; Fig. 6).

Spearman's rank correlation coefficient was calculated to test for correlations between the three dependent variables. We found a significant correlation between the number of looks and their duration ($rs = .82$, $p < .001$), while the other correlations were not

Varola et al. (2021), *PeerJ*, DOI 10.7717/peerj.12431

**Table 3  Summary of the mixed effect models and corresponding results.** The statistical models were applied first on the three seals together (top rows), and then excluding seal-1 from the group (bottom rows).

| | Response | Predictors | Estimate | SE | z | *p*-value |
|---|---|---|---|---|---|---|
| **All seals** | Number of looks | familiarity: control | −.19 | .22 | −.87 | .39 |
| | | Model: nr. of looks ∼ familiarity + (1 \| seal_id), class Glmer, family Poisson, link: log, $R^2$ conditional/marginal = .40/.02 | | | | |
| | Duration | familiarity: control | .05 | .03 | 1.49 | .14 |
| | | Model: duration ∼ familiarity + (1 \| seal_id), class: Glmer, family: Gamma, link: inverse, $R^2$ conditional/marginal = .002/.002 | | | | |
| | Latency | familiarity: control | −.02 | .03 | −.68 | .50 |
| | | Model: latency ∼ familiarity + (1 \| seal_id), class: Glmer, family: Gamma, link: inverse, $R^2$ conditional/marginal = .001/.000 | | | | |
| **Seals 2 and 3** | Number of looks | familiarity: control | −.36 | .24 | −1.51 | .13 |
| | | Model: nr. of looks ∼ familiarity + (1 \| seal_id), class Glmer, family Poisson, link: log, $R^2$ = NA/.148 | | | | |
| | Duration | familiarity: control | .05 | .03 | 1.46 | .15 |
| | | Model: duration ∼ familiarity + (1 \| seal_id), class: Glmer, family: Gamma, link: inverse, $R^2$ conditional/marginal = NA/ .002 | | | | |
| | Latency | familiarity: control | −.02 | .04 | −.52 | .60 |
| | | Model: latency_react ∼ familiarity + (1 \| seal_id), class: Glmer, family: Gamma, link: inverse, $R^2$ conditional/marginal = .001/.000 | | | | |

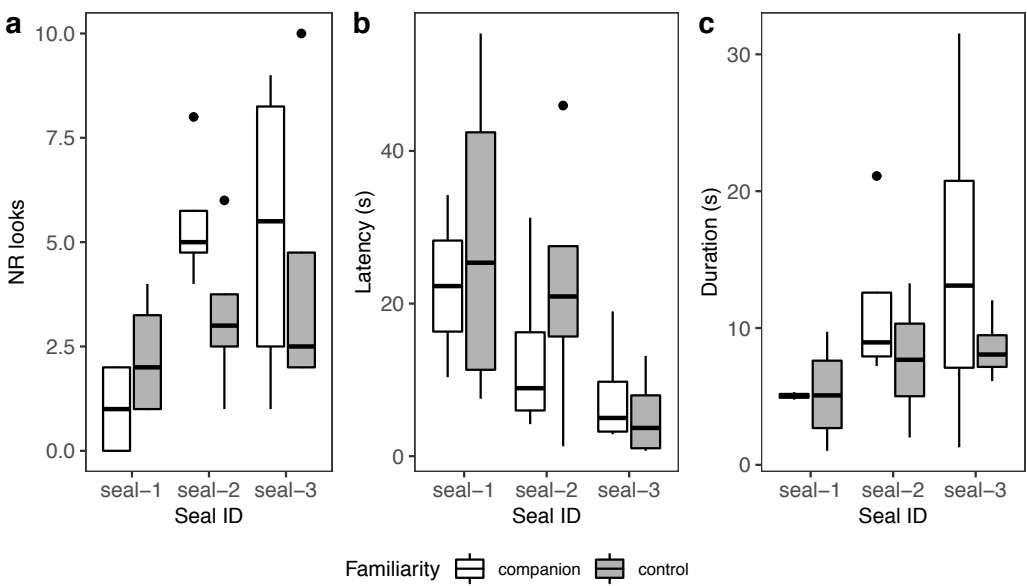

**Figure 4** **Graphic summary of the experimental results.** Boxplots showing the number of looks (A), their latency (B), and duration (C) for familiar (companion) relative to unfamiliar (control) blocks. For all variables, the box contains the interquartile range (*i.e.*, the range containing 50% of all observations) and the median value (middle line), while the whiskers represent values extending outside 50% of the observations.

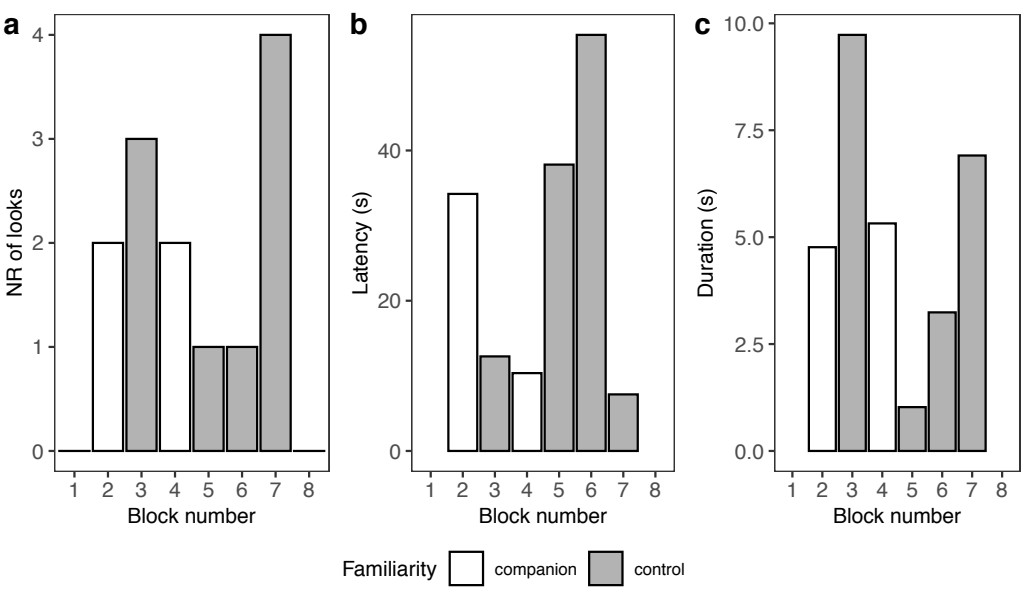

**Figure 5** **Details of seal-1 responses to playback.** Panels depict the number of looks (A), latency (B), and duration (C) for each block to either familiar (companion) or unfamiliar (control) blocks.

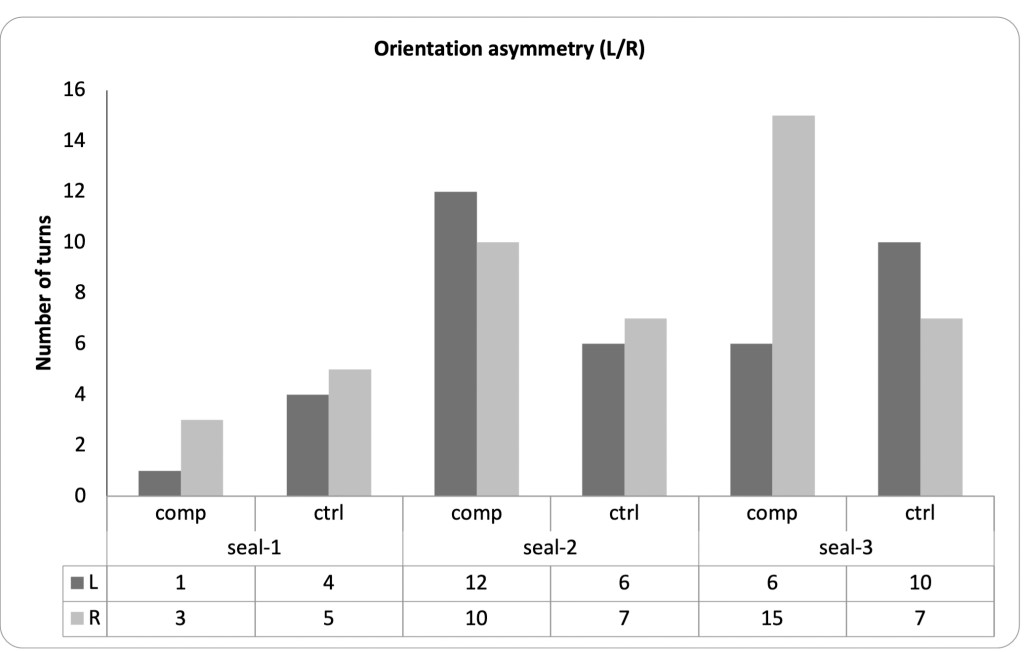

**Figure 6  Pattern of lateralization for the observed behaviors.** The graph depicts, for each seal, the direction of each head turn (*i.e.*, left or right) in response to either familiar (*i.e.*, companion) or unfamiliar (*i.e.*, control) vocalizations.

significant (duration and latency: rs $= -.41$, $p = .063$; latency and number of looks: rs $= -.40$, $p = .065$). These results were confirmed when excluding seal-1 from the analyses (number of looks and duration, rs $= 165.68$, $p < .001$; other ps $> .56$).

## Mixed-effects models

Generalized linear mixed-effects models were employed to statistically investigate behavioral differences between companion and control calls. GLMMs fitting a Poisson (for the number of looks) or gamma (for the duration and latency of reaction) model were used to predict the number of looks based on *familiarity condition* (fixed effect) while accounting for variability in each seal (random effect *seal ID*). No significant effects emerged from these analyses (Table 3).

Seal-1 was housed in different conditions and exposed to playbacks from multiple conspecifics. For this reason, we ran the models the second time after excluding data from this seal (*i.e.*, based on the data from seal-2 and seal-3). Even in this case, we found no difference between responses to companions and controls (Table 3).

## Principal component analysis

Number of looks, duration, and latency were fed into a PCA to identify a single measure summarizing the strength of response to playback. The first component accounted for 70% of the variance with an eigenvalue $> 1$ (loading coefficients: number of looks $= .64$, duration $= .62$, latency $= -.45$). A positive PC score indicates a strong response towards the sound source. These composite scores were fed into a linear mixed-effects model with
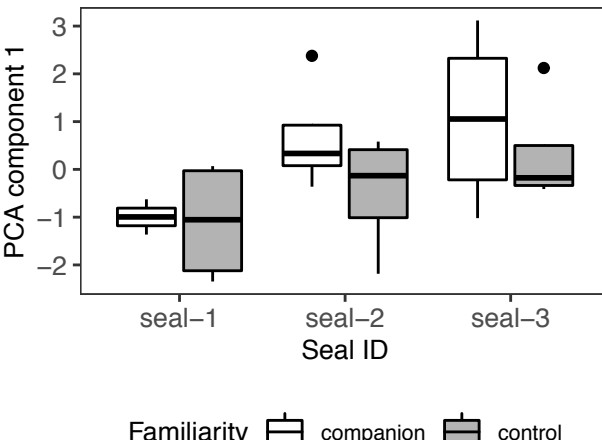

**Figure 7  PCA results.** Principal component scores (PC1) of seals' behavioural responses to calls from their companion and those from control (unfamiliar individuals). For all variables, the box contains the interquartile range (*i.e.*, the range containing 50% of all observations) and the median value (middle line), while the whiskers represent values extending outside 50% of the observations.

*familiarity condition* as fixed effect and *seal ID* as random effect (Shapiro-Wilk: W = .94, $p = .222$; model: pca $\sim$ familiarity condition + (1 | seal_id), class: lmer), with no significant outcome (familiarity condition: estimate = $-.79$; SE = .56, $t = -1.42$, $p = .17$; $r^2 = .26/.07$; Fig. 7).

Repeating the analyses without including seal-1 yielded comparable results. The first component accounted for 62% of the variance with an eigenvalue >1 (loading coefficients: number of looks = .68, duration = .66, latency = $-.32$). This composite score was fed into a linear mixed-effects model with *familiarity condition* as fixed effect and *seal ID* as random effect (Shapiro-Wilk: W = .94, $p = .315$; model: pca $\sim$ familiarity condition + (1 | seal_id), class: lmer), with no significant outcome (familiarity condition: estimate = $-.91$; SE = .66, $t = -1.37$, $p = .191$, $r^2 = \text{NA}/.112$).

## DISCUSSION

The current study set out to explore the ability of harbor seals to discriminate between familiar and unfamiliar vocalizations. We capitalized on a rare opportunity: three seals were admitted for rehabilitation twice in a relatively short time (approximately 6 months), first as orphaned pups and later as weaners affected by parasitic pneumonia. Our observations are compatible with the idea that the playback of conspecific calls may modulate seal behavior (*Tripovich et al., 2008*), although they are not supported by statistical analysis. However, since the present study is, to our knowledge, the first of its kind, it was limited by logistical difficulties due to the particular conditions of seal recruitment. Thus, our results require further investigation.

### Difference in response: seal-1 *vs* seal-2 and seal-3

Visual inspection of the graphs revealed that the first tested seal (seal-1) showed very little response to MACs compared to the other two seals and it seemed to respond more to unfamiliar vocalizations. This animal differed from the others for several reasons. First, during its stay at the Sealcentre, seal-1 shared an enclosure with many other pups, but physical contact with companions was prevented by fences. Conversely, seal-2 and seal-3 only had one companion each, with whom physical contact was possible. Second, seal-1 spent a shorter period with companions than seal-2 and seal-3 did (11–34 days in comparison to 45 and 48, see Table 1). The time elapsed between testing and seal-1's separation from its companions was also shorter than in the other two cases (only 190/191 days in comparison to 291 and 355). Third, seal-1, a male, was exposed to MACs belonging to pups of the opposite sex. In contrast, the other two seals were exposed to MACs belonging to pups of the same sex (male for seal-2 and female for seal-3). We speculated that especially the first two points may have played a key role in the reduced response shown by seal-1: by spending its first rehabilitation period with many other vocalizing pups, the animal was exposed to a more diverse stimulation coming from different sources. Moreover, being physically isolated from the other companions, seal-1 did not socialize with any of them, and thus it could not use visual or olfactory cues to reliably attribute calls to a given individual and reinforce individual vocal recognition. Taken together, these observations may justify a decreased chance to become familiar with the individual companions' vocalizations. Seal-1 and seal-2, both males, seemed to react faster to familiar as compared to unfamiliar calls, while for seal-3, a female, the response speed appeared to be similar in all conditions. We hypothesize that such a diverse response between the animals can be related to various aspects of pinnipeds' social structure: males generally display more active behavior than females from an early stage of life (*Bowen, 1991*; *Schusterman, 2008*) and become aggressive towards other males during the mating season (*Hayes et al., 2004*). Moreover, harbor seals show strong site fidelity throughout the year (*Cordes & Thompson, 2015*). Male seals able to discriminate between familiar and unfamiliar conspecifics would gain a strong advantage in terms of reproductive success and energy expenditure. The faster reaction shown by seal-1 and seal-2 towards familiar calls may be a sign of this ability. However, possibly due to the small sample size, we found no statistical evidence to reject the hypothesis of discrimination between call types. Future studies, considering aspects such as sex and number of familiar individual calls presented to each tested seal, might help clarifying these intriguing preliminary observations.

### Correlation between variables and head-turn asymmetry

We observed a significant correlation between the number of looks and their duration: seals responded longer to MACs which also elicited more looks. These results suggest that harbor seals do not respond equally to all conspecific vocalizations but despite that, unfortunately, we did not find any significant association between the seals' response and the type of MAC, whether familiar or not. Furthermore, we tested for possible response asymmetries: previous evidence consistently showed that the left-brain hemisphere is responsible for vocal production and perception in animals and humans

(*Ghazanfar, Smith-Rohrberg & Hauser, 2001*; *Siniscalchi et al., 2012*); accordingly, asymmetries in head orientation towards familiar and unfamiliar sounds have been found in pinnipeds as well as other animals, with a predominant right-ear (left brain hemisphere) advantage for familiar calls (*Böye, Güntürkün & Vauclair, 2005*). In pinnipeds, however, most of these studies do not discriminate between the familiarity of single individual calls; rather, they refer to all conspecific sounds as "familiar sounds". Since all playbacks in our study were conspecific calls, we focused on the difference between "truly" familiar seals (*i.e.*, former rehabilitation companions) and unfamiliar pups (*i.e.*, unknown seal pups with which the focal animal had no previous contact). We found no specific or overall statistical association. Several factors may contribute to this null result. First, this lack of association may be due to the small sample size. Second, it may reflect a true lack of lateralization. Third, it may relate to the fact that we only presented conspecifics' sounds, while a different result may have emerged if we had tested animals using heterospecific sounds as control stimuli. Nevertheless, this null result can inspire future research on head-asymmetry response to conspecifics *vs.* heterospecifics in harbor seals using a similar paradigm.

### Limitations and future directions

Given the novelty of the topic addressed in the current study, the experimental design had many limitations. First and foremost, due to the unique testing conditions, we were only able to investigate three animals. This low number certainly limited our inferential statistics and, consequently, our conclusions are mostly based on pure observations. Moreover, the difference in number of companions between the first seal and the other two mirrored the difference in results. To alleviate this shortcoming, future studies targeting long-term discrimination under similar circumstances (for example in seal rehabilitation centers) should consider recording vocalizations from as many seal pups as possible during their first rehabilitation period. This way, it would be possible to maximize the chances of obtaining usable data in the event of a second intake at the same center. This approach would be helpful in at least three ways. First, a larger sample size would enable testing for additional factors that may influence discrimination, such as the sex of the pup and/or of the recorded seal, and combinations thereof. Second, vocal discrimination in one animal may be influenced by the number of its companions, as suggested by the differences observed between seal-1 and the other two seals. Third, natural variability in the time passed between the first and the second admission might further inform on long-term discrimination capacities in harbor seals: is there a temporal gradient for familiar vocalizations? Do seals discriminate between conspecific vocalizations regardless of how much time has passed since they were heard? In this case, if we had had a larger sample size, we could have included elapsed time as a predictor of seal responsiveness to familiar voices in the model.

## CONCLUSIONS

The current study aimed at investigating, for the first time, the ability of harbor seals to discriminate between familiar and unfamiliar conspecifics during puppyhood. The

most parsimonious interpretation of our null results is that the seals do not possess this capacity. A possible reason for this absence of effect would be that conspecific calls were not biologically relevant and worth remembering. Alternative explanations for the absence of effect could be that our experiment was underpowered or not well-enough targeted to this species' ecology. Harbor seals are subject of increasing scientific work; empirical evidence on vocal learning, sound production, time perception, hearing, auditory memory, and social dynamics, to name a few, is growing (*Heinrich, Dehnhardt & Hanke, 2016*; *Ravignani et al., 2016*; *Kastelein, Helder-Hoek & Terhune, 2018*; *Wilson et al., 2018*; *Adams et al., 2020*; *Galatius et al., 2020*; *Heinrich, Ravignani & Hanke, 2020*; *Borda et al., 2021*). Once a clearer socioecological picture is available, we hope that our experiment will be modified to properly target the threshold of long-term memory capacities in harbor seals.

## ACKNOWLEDGEMENTS

The authors would like to thank all the veterinarians and seal nurses at the Sealcentre Pieterburen for their help and support in handling the seals and coordinating the experiment procedure in accordance with the releasing schedule. We are also grateful to all the researchers who recorded the original sounds used for playbacks, especially to Marianne de Heer Kloots, who helped us perform the experiment, Koen de Reus, who shared with us his study material, and Alice Lowry. We would also like to thank Jakub Wesecki for proofreading the manuscript.

### Funding

The work of Andrea Ravignani and Laura Verga was supported by a Max Planck Research Group (MPRG) awarded to Andrea Ravignani. The funders had no role in study design, data collection and analysis, decision to publish, or preparation of the manuscript.

### Grant Disclosures

The following grant information was disclosed by the authors:
Max Planck Research Group (MPRG).

### Competing Interests

The authors declare there are no competing interests.

### Author Contributions

- Mila Varola and Andrea Ravignani conceived and designed the experiments, performed the experiments, analyzed the data, prepared figures and/or tables, authored or reviewed drafts of the paper, and approved the final draft.
- Laura Verga analyzed the data, prepared figures and/or tables, authored or reviewed drafts of the paper, and approved the final draft.
- Marlene Gunda Ursel Sroka authored or reviewed drafts of the paper, and approved the final draft.

- Stella Villanueva authored or reviewed drafts of the paper, provided support on-site and feedback on the experiment's impact on the animals well being, and approved the final draft.
- Isabelle Charrier conceived and designed the experiments, authored or reviewed drafts of the paper, and approved the final draft.

## Animal Ethics

The following information was supplied relating to ethical approvals (*i.e.*, approving body and any reference numbers):

Sealcentre Pieterburen

## Data Availability

The playback annotations are available in the Supplementary File.

## Supplemental Information

Supplemental information for this article can be found online at http://dx.doi.org/10.7717/peerj.12431#supplemental-information.

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
