# Peer review of "Can harbor seals (Phoca vitulina) discriminate familiar conspecific calls after long periods of separation?"

_PeerJ, doi:10.7717/peerj.12431_

## Round 0.1 · original submission · Minor Revisions

As you can see, both Reviewers appreciate your work and only asked for some minor amendments. Please revise the paper accordingly and I would be happy to reconsider it.

·

Basic reporting

This is a well-conducted study examining the ability of harbour seals to recognise vocalisations of pups with which they had been raised versus the vocalisations of unfamiliar pups. They were tested on re-capture some time after release and hence after exposure to the vocalizations of other pups. Probably due to the low sample size (n=3) the results were not significant.

The English expression is very good.

Experimental design

Realising the low probability of re-capture of the same individuals for which recorded vocalisations were available, I recommend publication of this study but suggest that the text be shortened to some degree. This should be possible because there is repetition at different places in the manuscript and it would allow some more figures to be added, as suggested below.
Some important details are missing from the Methods (see below).

Validity of the findings

45-47: This reference to memory span and speech rate in humans is not particularly relevant and should be deleted.

57: Here it is suggested that long-term memory recognition could be ‘an evolutionary by-product of strong selective pressures’. This doesn’t make much sense unless you specify what those selective pressures may be.

124-127: Say exactly how old each seal was at the time of re-capture and also at the time of testing the responses to playback of the vocalisations. This should be in the text as well as in the Table.

189-220: It would be helpful to include some images from the videotapes showing the behaviour scored. Also, it would be valuable to present some sound spectrograms of the calls presented in the playbacks.

191-193: This description of the visual fields is not very clear. Perhaps adding diagrams of the binocular and monocular visual fields in the horizontal and vertical planes would help the reader.

208-209: Because I think that seals have a great deal of mobility of the head, the plane of the >900 head turning should be stated.

Footnote to Figures 3 and 5: Say exactly what is plotted (e.g. medians and type of distribution).

Error of grammar
49: Change ‘showed’ to ‘have shown’.

Additional comments

Please see above.

·

Basic reporting

Varola et al. tested the hypothesis that harbor seal pups can recognize familiar from unfamiliar conspecific sounds after a long separation.

The writing is clear, the topic is well introduced.

Experimental design

Varola et al. took advantage of having 3 seals admitted for rehabilitation twice in 6 months (first as pups and then as weaners). The seals were exposed to previously recorded sounds of familiar and unfamiliar conspecifics during the second rehabilitation period.

The experimental question is well-defined and interesting.

Validity of the findings

The behavioural responses measured (lateralization patterns, number of looks, latency before looking and duration of looks) yield no sign of recognition between familiar and unfamiliar sounds. Yet, the authors cautiously argue that the data visualization could support the idea that seals differentiate familiar and unfamiliar conspecifics. However, together with the low sample size, several differences between the tested animals (sex, separation duration, rearing conditions, etc.) are present and make the result interpretation very difficult. Still, the authors reported their results with complete transparency and discussed them adequately.

Additional comments

I do not have concerns with this article and think it is worth being published as it is. I have a few comments and text issues (see later) that the authors can consider.

While the authors clearly state their hypothesis about the head turns direction for familiar calls (right turns expected), their assumptions about the other behavioural measurements are pretty unclear to me. For example, should we expect lower or higher latency to respond to familiar compared to unfamiliar calls? Ultimately, it does not make much of a difference. What is important here is whether there are behavioural differences between familiar and unfamiliar calls, but discussing your hypotheses for all behaviours measured beforehand (if you had some) could be entertaining for the readers.

The principal component analysis was new to me. Can this approach be problematic when combining behaviours for which you expect opposite direction? For example: for an animal that reacts to a call, I would expect a higher number of head turns and lower latency to respond than an animal that is not responding. Isn’t there a higher risk of information loss in using a PCA in this case?

About the lateralization pattern: Although the authors did not find any left-right correlations with the behavioural responses, it could be interesting to represent the lateralization patterns for all the recorded behaviours graphically.

Small issues:
1. In the last paragraph of the section ‘Study site and animals’ (l.136 to l.141), the age of the animals is described for seal-1 and seal-3 as stated but not for seal-2. The authors might want to add its age in parenthesis.
2. L.179: there might be something wrong with the reference (see ‘t)

---

## Round 0.2 · Minor Revisions

After consulting with the Section Editor, we would like to ask the authors to incorporate in the paper the following changes:

A: The authors make the title better aligned with what they could actually test and found, something like "Rehabilitated seals may not discriminate familiar and unfamiliar conspecific calls after long periods of separation." because all they can speak to is discrimination or preference for familarity generally, not memory or recognition of individuals specifically. The authors even address this in relation to previous findings in the discussion.

B. The authors change sentences such as lines 388-389 and some parts of the introduction that aim to test memory for individuals.

---

## Round 0.3 · accepted · Accept

It seems to me that the authors addressed adequately all reviewers and editorial comments, and I am therefore happy to accept the paper in this final version.